# Ultrasound-guided puncture drainage versus surgical incision drainage for deep neck space abscesses: a protocol for a systematic review with meta-analysis and trial sequential analysis

Wei Nie,[1] Li Du,[2] Guo Chen,[1] Yongbo Zheng,[3] Xibiao Yang,[4] Bo Li,[3] Weiyi Zhang,[1] Jianqiao Zheng  [1]

**Correspondence to**
Dr Jianqiao Zheng;
zhjq1983@163.com

## ABSTRACT

**Introduction** Deep neck space abscesses (DNAs) are serious surgical emergencies, associated with life-threatening complications. Surgical incision and drainage combined with antibiotics is the main treatment for DNAs, but drawbacks still exist. Ultrasound-guided puncture drainage is an alternative treatment for some DNAs with limited clinical evidence. Hence, the optimal drainage technique for the treatment of DNAs remains unclear. Therefore, we will perform a protocol for a systematic review and meta-analysis to identify the efficacy of ultrasound-guided puncture drainage for DNAs.

**Methods and analysis** PubMed, Ovid Medline, Cochrane Library, Embase, Web of Science, China National Knowledge Infrastructure, Wanfang database, VIP database and trial registry databases will be searched from inception to September 2023 to identify randomised controlled trials of patients diagnosed with DNAs accepting ultrasound-guided puncture drainage. The primary outcome will be the length of hospital stay. The secondary outcomes will be the cure rate, incidence of retreatment, complications and overall cost to the healthcare system. Fixed-effects or random-effects model will be used according to the statistical heterogeneity. Mean differences or standardised mean differences with 95% CIs for continuous data and risk ratio (RR) with 95% CIs for dichotomous data. The Cochrane risk-of-bias tool 2, Grading of Recommendations Assessment, Development and Evaluation (GRADE) and trial sequential analysis will be conducted to evaluate the evidence quality and control the random errors. Funnel plots and Egger's regression test will be performed to evaluate publication bias.

**Ethics and dissemination** Ethical approval was not required for this systematic review protocol. The results will be disseminated through peer-reviewed publications.

**PROSPERO registration number** CRD42023441031.

## INTRODUCTION

Deep neck space abscesses (DNAs) are serious otolaryngology-head and neck surgery emergencies, defined as bacterial infections in the potential spaces and fascial planes of the neck, which could result in significant morbidity with potential mortality.[1–4] Dental infection is the most common cause of DNAs, and the prevalence of DNAs has been reduced with the advent of broad-spectrum antibiotics.[5] However, DNAs remain serious, spread rapidly and are associated with life-threatening complications.[6–9] Common and potentially life-threatening complications range from 15% to 35% in patients with DNAs,[5 10 11] including airway compromise, descending mediastinitis, pericarditis, pneumonia, pleural empyema, carotid arterial erosion, meningitis, sepsis, respiratory distress, jugular vein thrombosis, disseminated intravascular coagulation and extracranial or intracranial extension of infection.[5 8–13] Therefore, the early detection of abscesses and thorough treatment are crucial.

Traditionally, the main treatment method for DNAs is complete surgical incision and drainage with appropriate administration of antibiotics.[14–19] However, surgical drainage has several drawbacks. First, the risk of inadvertent injuries to neighbouring vital structures (including the internal jugular vein, carotid artery and cranial nerves)

is associated with devastating complications. Second, surgical trauma danger is increased as the inflammatory anatomy tissue is obscure and easily bled during incision when the abscess is deep. Third, patients suffering from surgical trauma stress become frail and more sensitive to pathogenic bacteria in the environment, increasing the risk of infection-related complications. Fourth, tumour-related DNAs could cause tumour dissemination during open surgical incision and drainage. Fifth, surgical incision and drainage was performed under general anaesthesia in most circumstances, which will increase the anaesthesia-related risk, especially in patients with poor medical condition DNAs. Finally, surgical incision scars increase the risk of incision disunion and cause aesthetic defects. Therefore, a non-invasive, effective alternative to open surgical incision drainage may be beneficial to patients with DNAs.

Minimally invasive techniques, such as ultrasound-guided puncture drainage, have been used to treat abscesses of the head and neck, even some DNAs.[20–37] Ultrasound-guided puncture drainage can avoid the above drawbacks of surgical incision drainage and provide several advantages. First, ultrasound-guided puncture drainage can accurately locate the DNAs, eliminating trauma to the neighbouring vital structures and avoiding tumour dissemination during surgical incision. Second, the drainage tube can be placed under the real-time guidance of ultrasound, which is beneficial for thoroughly draining the abscess. Third, the real-time capability of ultrasound could facilitate dynamic observation after drainage and monitoring the adequacy of drainage. Finally, ultrasound-guided puncture drainage does not cause skin non-union or aesthetic defects. Several studies have shown that ultrasound-guided puncture drainage is a more effective option than surgical incision drainage for some DNAs.[20–36] In addition, ultrasound-guided puncture drainage can reduce the length of hospital stay.[25–29] However, most of these studies were case series, reviews and retrospective studies, and prospective and randomised controlled trials are rare.[20-36] In this situation, controversies still exist in the selection of drainage technique for the treatment of DNAs.[37 38]

Therefore, it is necessary to conduct a systematic review and meta-analysis to analyse the clinical efficacy of ultrasound-guided puncture drainage for DNAs. The outcomes of this systematic review will provide evidence for better clinical decision-making and possible future directions for further clinical trials.

## Objectives

We are performing this systematic review protocol with meta-analysis and trial sequential analysis (TSA) of randomised clinical trials to evaluate the clinical efficacy and safety of ultrasound-guided puncture drainage for DNAs.

## METHODS AND ANALYSIS
### Design and registration of the review

We reported this protocol according to the Preferred Reporting Items for Systematic Reviews and Meta-Analyses Protocols (PRISMA-P) guidelines and have registered the protocol with PROSPERO 2023 (registration number: CRD42023441031).[39] We will perform this systematic review and meta-analysis based on the Cochrane Handbook and report the results following the PRISMA statement.[40 41] This study is anticipated to begin searching in September 2023 and will be completed in March 2024.

### Patient and public involvement statement

Patients or the public were not involved in the design, conduct, reporting or dissemination plans of our research.

### Inclusion criteria for study selection
#### Types of studies

Only randomised controlled trials (RCTs) involving the clinical efficacy of ultrasound-guided puncture drainage for DNAs will be included. There will be no language restrictions.

The exclusion criteria were as follows: (1) studies comparing ultrasound-guided needle puncture aspiration and ultrasound-guided catheter drainage or studies comparing ultrasound-guided needle puncture aspiration with other minimally invasive techniques (CT-guided needle puncture aspiration); (2) studies with data that could not be used for statistical analysis after data transformation, or studies with incomplete data, or original data that could not be obtained after contacting the corresponding authors; and (3) duplicate publications, letters or editorials, abstracts from conferences and reviews.

#### Types of participants

Participants diagnosed with DNAs who accept ultrasound-guided puncture drainage will be included. There will be no limitations on participants' age, gender, ethnicity, body mass index (BMI) or American Society of Anesthesiologists (ASA) classification.

#### Types of interventions/controls

The intervention group will be the participants with DNAs accepting any kind of ultrasound-guided puncture drainage, while the control group will receive any kind of surgical incision drainage for DNAs.

#### Types of outcome measures
*Primary outcomes*

The primary outcome will be the length of hospital stay.

*Secondary outcomes*

1. The cure rate.
2. Incidence of retreatment (including secondary puncture drainage or incision drainage).
3. Incidence of complications: any kind of complications will be included. Complication criteria included drainage-related mechanical injury (vascular injury,

nerve injury, airway injury and bleeding), infectious complications (sepsis, mediastinal infection, pericarditis, pneumonia, arterial infection), airway obstruction, jugular vein thrombosis, skin non-union and other puncture complications.

4. Overall cost to the healthcare system.

## Search strategy

Two reviewers (WN and LD) will independently perform the search, and any disagreements will be arbitrated by a third reviewer (WZ). English and Chinese electronic databases will be searched from inception to September 2023 for published literature. English databases (including PubMed, Ovid Medline, Cochrane Library, Embase and Web of Science) and Chinese databases (China National Knowledge Infrastructure (CNKI), Wanfang database and VIP Database) will be searched. In addition, reference lists of each study and the trial registry database (Clinical Trials.gov, WHO International Clinical Trials Registry Platform and Chinese Clinical Trials Registry) will also be scrutinised as to avoid missing studies and ongoing or unpublished clinical trials.

The following search terms will be used in the search strategy: deep neck space abscesses, ultrasound, surgical and randomised controlled trial. Related search terms will also be translated into Chinese for literature research and study identification in Chinese databases. The search strategies are listed in online supplemental Appendix file 1. To avoid missing published studies during the systematic review preparation, we will update the literature search results prior to the final publication of systematic reviews.

## Data collection and analysis

### Selection of studies

Two authors (WN and LD) will independently screen the potentially eligible studies by reading titles and abstracts. All identified and relevant full-text publications will be retrieved from the database search process by screening the full text thoroughly, obviously ineligible records will be eliminated, and the reasons for exclusion will be recorded. Any disagreement on eligibility of studies will be resolved through discussion or arbitrated by a third review coauthor (JZ). A fourth reviewer (WZ) will check out all procedures carefully prior to the final confirmation of the data extraction. Duplicate publications and companion papers of the same trial will be discussed by all review authors. The entire search process is displayed in the PRISMA flow diagram (figure 1).

### Data extraction

Two review authors (WN and LD) will extract data from all included studies according to a standardised data collection form (Excel version 2013, Microsoft, Washington DC, USA). The data extraction form included the publication information (the first author's name, author's country publication year), participants' demographic data (sample size; age; gender; BMI; ASA physical status classification levels; inclusion and exclusion criteria if necessary), locations of DNAs, experience of the operator, detailed information on drainage techniques (type of drainage techniques: surgical incision drainage or ultrasound-guided puncture drainage; days of drainage tube retention; and retreatment of multiple punctures or incisions), microbial cultures, antibiotic treatment(s) and any kind of outcomes including primary and secondary outcomes.

Study design characteristics, including the following domains, randomisation, allocation concealment, blinding (patients, treatment providers, outcome investigators), incomplete outcome data collection and statistical analysis, and outcome reporting, will be recorded. Continuous data will be recorded as the mean±SD, and dichotomous data will be the recorded as percentages or proportions. A third review author (GC) will cross-check the data to ensure precision. We will contact the corresponding author of the research via email to acquire the missing or incomplete data for analysis as much as possible. In addition, the record will be excluded from the final review if no reply is received. Numerical data in the graphs will be extracted by Adobe Photoshop if necessary.[42] In addition, mean and SD will be estimated from the median, mid-range and/or mid-quartile range as described by Luo *et al* and Wan *et al*.[43 44]

### Quality assessment

Risk of bias in each included study will be evaluated independently by two review authors (YZ and WZ) under the guidance of the Cochrane risk-of-bias tool 2 (RoB V.2.0).[45] The methodology including random sequence generation, allocation concealment, blinding (blinding of participants, personnel and outcome assessment), incomplete outcome data, selective outcome reporting, other risks of bias and overall risk of bias will be assessed. Each included study will be evaluated by the Cochrane risk-of-bias tool and then classified into three levels (low risk, unclear risk and high risk of bias).[40 46] Any discrepancies will be settled through discussions by all review authors or arbitration by a third reviewer (GC). The assessment of risk of bias is listed in online supplemental Appendix file 2.

Grading of Recommendations Assessment, Development and Evaluation (GRADE) methodology will be used to assess the quality of the evidence for each outcome.[47] The quality of effect estimates will be classified as high, moderate, low or very low depending on the risk of bias, consistency, directness, precision, publication bias, large effect and plausible confounding.[48] After the assessment, a table concerning the GRADE evidence profile will be created using GRADE profiler V.3.6.1 to rate the quality of all outcomes.

### Measures of treatment effect

Mean differences (MDs) with 95% CIs and standardised mean differences (SMDs) with 95% CIs will be used for continuous outcome data with the same unit and different units, respectively. The relative risks (RRs) with 95% CIs

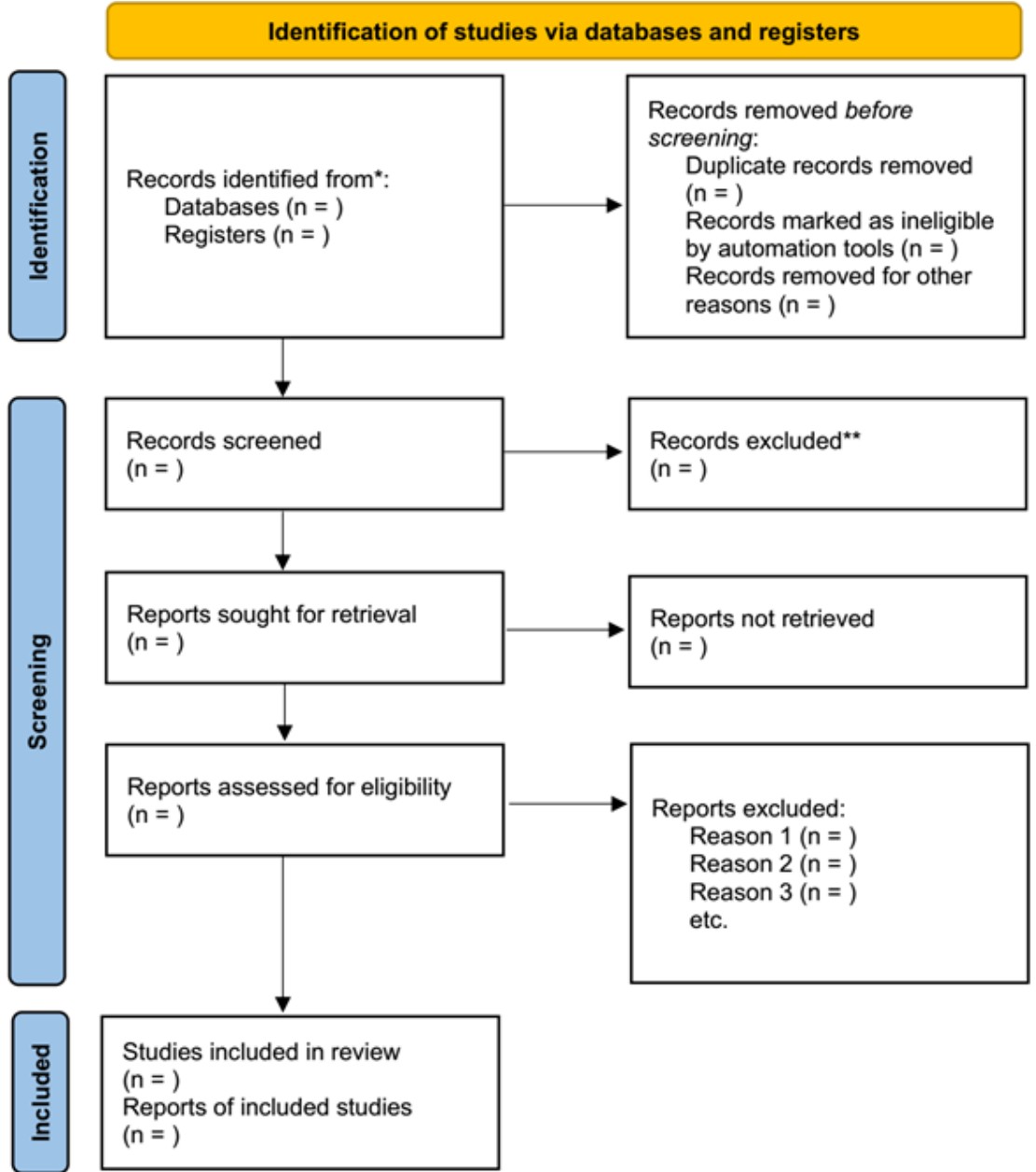

**Figure 1** The PRISMA flow diagram. PRISMA, Preferred Reporting Items for Systematic Reviews and Meta-Analysis. *Consider, if feasible to do so, reporting the number of records identified from each database or register searched (rather than the total number across all databases/registers). **If automation tools were used, indicate how many records were excluded by a human and how many were excluded by automation tools.

will be used for dichotomous outcome data. Heterogeneity analyses will be performed by Review Manager V.5.4 (RevMan, Cochrane Collaboration, Oxford, UK). Statistical heterogeneity will be assessed by the standard $\chi^2$ test ($\alpha=0.1$) and $I^2$ test. If $p \geq 0.1$ and if $I^2 \leq 50\%$, the fixed-effects model will be used. If $p < 0.1$ or $I^2 > 50\%$, the random-effects model will be applied.[40] A p value <0.05 will be considered statistically significant.

### Trial sequential analysis

The required information size (RIS) will be calculated to correct the risks of random errors by trial sequential analysis (TSA) using the TSA programme V.0.9.5.10 Beta (Copenhagen Trial Unit, Copenhagen, Denmark).[48–50] TSA programme version is available at http://www.ctu.dk/tsa.[51] Each outcome will be detected by the RIS, the cumulative Z-curve and the TSA monitoring boundaries.[52 53]

To calculate the RIS, the risk of type I error will be maintained at 5% with a power of 90% in both continuous and dichotomous outcomes.[54] For continuous outcomes, a mean difference of the observed SD/2 will be used to calculate the RIS. For dichotomous outcomes, the proportion or percentage from the control group, a relative risk reduction (RRR) of 20% or −20% will be used

to calculate the RIS. When there is limited evidence available about the intervention under investigation, we will estimate a clinically relevant intervention effect by using clinical experience and evidence from related areas.[53–55]

## Subgroup analysis

The results will be comprehensively interpreted through an analysis of subgroups or subsets as much as possible. If sufficient trials are available, data from different participants' ages and different locations of DNAs will be analysed independently.

► Different participants' ages (infants, children, adults and aged).
► Different locations of DNAs (parapharyngeal space, submandibular space, masseter space, retropharyngeal space, sublingual space, prevertebral space and carotid space).

The interaction p value will be used to test the statistically significant subgroup difference. If a significant difference between subgroups exists (testing for interaction $p<0.05$), the results for individual subgroups will be reported separately.[40]

## Assessment of publication biases

Visual judgement of the funnel plot asymmetry and Egger's regression test will be performed to estimate the potential publication bias, while more than 10 original studies involved the same outcome measure.[56 57] Analyses of reporting bias will be performed by Stata/MP 16.0 (StataCorp, College Station, Texas, USA). The effect sizes of each included study will be normally symmetrically distributed around the centre of a funnel plot in the absence of publication bias.[56] In the case of publication bias, trim and fill analysis will be performed to estimate the number of missing studies that might exist and amend the bias.[58] A level of $p<0.05$ will be considered statistically significant and indicated potential publication bias.

## Grading the quality of evidence

The quality of evidence for each outcome will be assessed using the GRADE criteria.[47] The quality of effect estimates will be classified as high, moderate, low or very low depending on the risk of bias, consistency, directness, precision and publication bias.[59] Data from randomised controlled trials will be classified as high-quality evidence according to GRADE. However, it can be degraded according to risk of bias, imprecision, inconsistency, indirectness or publication bias.

## Timelines

Formal screening of search results will begin in September 2023. Data extraction will begin in December 2023. The project will be completed in March 2024.

## Author affiliations
[1]Department of Anesthesiology, West China Hospital, Sichuan University, Chengdu, Sichuan, China
[2]Department of Anesthesiology, Sichuan Cancer Hospital and Research Institute, Chengdu, Sichuan, China
[3]Department of Otorhinolaryngology, Head and Neck Surgery, West China Hospital, Sichuan University, Chengdu, Sichuan, China
[4]Department of Radiology, West China Hospital, Sichuan University, Chengdu, Sichuan, China

**Contributors** JZ and LD conceived the idea for this systematic review. All authors (WN, LD, GC, YZ, XY, BL, WZ and JZ) developed the methodology for the systematic review. The manuscript was drafted by JZ and GC and revised by all authors. WN and LD will screen potential studies and extract the data. YZ and WZ will undertake risk-of-bias assessment and assess the evidence quality. WN and JZ will conduct the data synthesis. All authors contributed to the research and agreed to be responsible for all aspects of the work.

**Funding** The authors have not declared a specific grant for this research from any funding agency in the public, commercial or not-for-profit sectors.

**Competing interests** None declared.

**Patient and public involvement** Patients and/or the public were not involved in the design, or conduct, or reporting, or dissemination plans of this research.

**Patient consent for publication** Not applicable.

**Provenance and peer review** Not commissioned; externally peer reviewed.

**ORCID iD**
Jianqiao Zheng http://orcid.org/0000-0002-8091-1837

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
