## [Reviewer comments · BMJ Open]

ARTICLE DETAILS

TITLE (PROVISIONAL)	Ultrasound-guided puncture drainage vs surgical incision drainage for deep neck space abscesses: a protocol for a systematic review with meta-analysis and trial sequential analysis
AUTHORS	Nie, Wei; Du, Li; Chen, Guo; Zheng, Yongbo; Yang, Xibiao; Li, Bo; Zhang, Weiyi; Zheng, Jianqiao

VERSION 1 – REVIEW

REVIEWER	Arslan, Hande Ministry of Health Samsun Education and Research Hospital, ENT
REVIEW RETURNED	24-Jul-2023

GENERAL COMMENTS	The method and statistical method of the study were constructed in accordance with the rules, as it should be. The only point I can criticize here is that there is only one otolaryngologist among the authors. In my opinion, at least one otolaryngologist should be added to the study to check the suitability of the articles. Even adding a radiologist would make the study more valuable.
--

REVIEWER	Blanco, Luis Sección de Cirugía de Cabeza y Cuello, Cirugia
REVIEW RETURNED	19-Sep-2023

GENERAL COMMENTS	Congratulate the authors for the initiative due to the lack of consensus on treatment. In addition, I would like to know if they consider puncture drainage or drainage with placement of a cutaneous catheter. Also a limitation of the efficacy variables that modify the days of hospitalization are the comorbidities and intercurrents of the patients that prolong the stays without having a direct relationship with the proposed treatment. In addition to these two clarifications, I congratulate you again.
---

REVIEWER	Tsikopoulos, Alexios Aristotle University of Thessaloniki, Medicine
REVIEW RETURNED	26-Nov-2023

GENERAL COMMENTS	Generally, this systematic review and meta-analysis investigates a debatable clinical practice, shedding light to a relatively new method of treatment of deep neck space abscesses. It is to my knowledge, the first meta-analysis comparing surgical treatment with ultrasound-guided drainage, and therefore it is definitely worth of publishing.
---

	The method of the execution of this analysis is well documented and according to the guidelines, since it was based on the Cochrane Handbook and reported the results following the PRISMA statement. The statistics are valid. Additionally, the fact that it incorporates uniquely RCTs adds strength to the results. I agree that the heterogeneity is expected due to the variation of the experience of operators varied in the ultrasound-guided drainage and surgical incision drainage techniques. The use of english is more than acceptable.
--	--

VERSION 1 – AUTHOR RESPONSE

Reviewer: 1

The only point I can criticize here is that there is only one otolaryngologist among the authors. In my opinion, at least one otolaryngologist should be added to the study to check the suitability of the articles.

Agree and Revised. After affirming the clinical value of this research, Professor Bo Li, another otolaryngologist agreed to join our team, and check the suitability of the articles.

Even adding a radiologist would make the study more valuable.

Agreed and Revised. Professor Xibiao Yang, a radiologist has also agreed to join our team to evaluate the diagnosis of deep neck abscesses in the included articles, with the aim of assisting us in evaluating the quality of studies included in the review.

Reviewer: 2

In addition, I would like to know if they consider puncture drainage or drainage with placement of a cutaneous catheter.

This issue you have considered are of great help to us. Any kind of ultrasound-guided drainage techniques will be considered, including studies that meet the criteria for ultrasound-guided percutaneous drainage techniques. Studies that use ultrasound guidance for percutaneous puncture drainage or drainage with the placement of a cutaneous catheter, provided that they meet our criteria, will also be considered.

A limitation of the efficacy variables that modify the days of hospitalization are the comorbidities and intercurrents of the patients that prolong the stays without having a direct relationship with the proposed treatment.

Thank you very much for your suggestions. Considering the significance of clinical treatment outcomes, hospital stay can be used as the primary outcome, and indeed, there are studies that have used hospital stay as an important research indicator.¹⁻⁴ In addition, ultrasound-guided puncture drainage can reduce the length of hospital stay.¹⁻⁴ Therefore, we have chosen hospital stay as the primary research indicator in this study.

1. Biron VL, et al. Surgical vs ultrasound-guided drainage of deep neck space abscesses: a randomized controlled trial: surgical vs ultrasound drainage. *J Otolaryngol Head Neck Surg* 2013; 42:18.

2. Dabirmoghaddam P, et al. Is ultrasonography-guided drainage a safe and effective alternative to incision and drainage for deep neck space abscesses? *J Laryngol Otol* 2017; 131:259-263.

3. Limardo A, et al. Ultrasound-guided drainage vs surgical drainage of deep neck space abscesses: A randomized controlled trial. *Acta Otorrinolaringol Esp (Engl Ed)* 2022; 73:4-10.

4. Fan X, Tao S. Comparison of ultrasound-guided puncture drainage and incision drainage for deep neck abscess. *Gland Surg* 2021;10: 1431-1438.

5. Strassen U, et al. Ultrasound-Guided Needle Aspiration vs. Surgical Incision of Parotid Abscesses. *J Clin Med* 2022; 11:7425.

Reviewer: 3

Comments to the Author: I agree that the heterogeneity is expected due to the variation of the experience of operators varied in the ultrasound-guided drainage and surgical incision drainage techniques.

Yes, we agree with your opinion. If data are available, subgroup analyses will be performed to detect the heterogeneity due to the variation in the experience of operators who perform the ultrasound-guided drainage and surgical incision drainage techniques.